# Targeting Glutaminolysis Shows Efficacy in Both Prednisolone-Sensitive and in Metabolically Rewired Prednisolone-Resistant B-Cell Childhood Acute Lymphoblastic Leukaemia Cells

**DOI:** 10.3390/ijms24043378

**Published:** 2023-02-08

**Authors:** Yordan Sbirkov, Bozhidar Vergov, Vasil Dzharov, Tino Schenk, Kevin Petrie, Victoria Sarafian

**Affiliations:** 1Department of Medical Biology, Medical University of Plovdiv, 4000 Plovdiv, Bulgaria; 2Research Institute at Medical University of Plovdiv (RIMU), 4000 Plovdiv, Bulgaria; 3Department of Hematology and Medical Oncology, Clinic of Internal Medicine II, Jena University Hospital, 07743 Jena, Germany; 4Institute of Molecular Cell Biology, Jena University Hospital, 07745 Jena, Germany; 5School of Medicine, Faculty of Health Sciences and Wellbeing, University of Sunderland, Sunderland SR1 3SD, UK

**Keywords:** childhood ALL, metabolism, metabolomics, glutaminolysis, glutamine, EGCG (epigallocatechin gallate), V-9302, glucocorticoids, prednisolone, resistance

## Abstract

The prognosis for patients with relapsed childhood acute lymphoblastic leukaemia (cALL) remains poor. The main reason for treatment failure is drug resistance, most commonly to glucocorticoids (GCs). The molecular differences between prednisolone-sensitive and -resistant lymphoblasts are not well-studied, thereby precluding the development of novel and targeted therapies. Therefore, the aim of this work was to elucidate at least some aspects of the molecular differences between matched pairs of GC-sensitive and -resistant cell lines. To address this, we carried out an integrated transcriptomic and metabolomic analysis, which revealed that lack of response to prednisolone may be underpinned by alterations in oxidative phosphorylation, glycolysis, amino acid, pyruvate and nucleotide biosynthesis, as well as activation of mTORC1 and MYC signalling, which are also known to control cell metabolism. In an attempt to explore the potential therapeutic effect of inhibiting one of the hits from our analysis, we targeted the glutamine-glutamate-α-ketoglutarate axis by three different strategies, all of which impaired mitochondrial respiration and ATP production and induced apoptosis. Thereby, we report that prednisolone resistance may be accompanied by considerable rewiring of transcriptional and biosynthesis programs. Among other druggable targets that were identified in this study, inhibition of glutamine metabolism presents a potential therapeutic approach in GC-sensitive, but more importantly, in GC-resistant cALL cells. Lastly, these findings may be clinically relevant in the context of relapse—in publicly available datasets, we found gene expression patterns suggesting that in vivo drug resistance is characterised by similar metabolic dysregulation to what we found in our in vitro model.

## 1. Introduction

Childhood acute lymphoblastic leukaemia (cALL) is the most common type of paediatric cancer, but is also one of the most curable malignancies, with a 5-year overall survival (OS) rate of about 80% for T-cell ALL (~10–15% of cases) and above 90% for B-cell ALL (~80% of cases) [1]. Nevertheless, there is considerable inter- and intra-tumour heterogeneity, evidenced by the distinction of more than 20 genetically different moieties of B-ALL [2]. Thereby, about 10% of B-cell ALL cases relapse, and in those cases, five-year event-free survival, especially in certain moieties like with *IKZF* deletion, can fall as low as 30% [3]. Relapses are caused by insufficient drug response; resistance to glucocorticoids (GCs) is believed to be the most common reason for treatment failure [4]. Importantly, GCs, as well as several other standard chemotherapeutics such as L-asparaginase and methotrexate, exert their anti-leukaemic effect through modulation of cell metabolism and depletion of nutrients and precursors that are critical for the quickly dividing lymphoblasts [5]. There are a variety of molecular lesions that can lead to GC resistance: mutations in the GC receptor, downregulation of its expression, increased glycolysis or TCA (tricarboxylic acid) cycle, activation of RAS and NOTCH signalling and others [5]. There are also a number of studies which have looked at gene expression differences between matched samples at diagnosis and at relapse. Mostly, and not surprisingly, the findings of such analyses show changes in cell cycle, DNA replication and self-renewal [6,7,8,9]. 

Notably, very few links to cell metabolism have been suggested, even by meta-analysis [10]. Only the upregulation of carbohydrate metabolism genes [7] and the identification of four transcription factors frequently inactivated in B-ALL patients were suggested as metabolic “gatekeepers” regulating glycolysis and the TCA cycle [11]. To date, the transcriptomic data focusing on prednisolone resistance are scarce, and there are no proteomics or metabolomics studies on the functional status of cALL cells that have become resistant to GCs. Therefore, due to the importance of this class of chemotherapeutics and the need for better understanding of treatment failure and identification of novel drug targets, we have previously established an in vitro experimental system for studying relapse. We developed a prednisolone-resistant cell line (Sup-PR), which is a sub-clone of Sup-B15 cALL cells [12]. Having this unlimited source of what would mimic matched samples at diagnosis and relapse at hand, we hypothesised that since prednisolone exerts its function through targeting cell metabolism, then resistant cells may develop escape/rescue mechanisms to cope with certain metabolic challenges. 

Therefore, the main aim of our study was to investigate whether acquired GC resistance through the applied selective pressure in our in vitro model system (mimicking chemotherapeutic regimens in patients) would lead to the rewiring of gene expression and cells’ anabolic and catabolic processes towards a state of enhanced metabolic plasticity or “fitness”. After defining a number of dysregulated bioenergetic and signalling pathways in GC-resistant cells, we also wanted to exploit one of these “hits” (glutaminolysis) as a potential therapeutic target. 

## 2. Results

### 2.1. Gene Expression and Metabolomics Analyses of Prednisolone-Resistant Cells Suggest Dysregulated Glycolysis, Amino Acid and Nucleotide Metabolism

In this study, we aimed to elucidate the molecular changes in GC-resistant cells that may, on the one hand, contribute to the lack of chemotherapy response, and on the other, may present druggable therapeutic targets. We first carried out RNA-sequencing of these matched prednisolone-sensitive and -resistant cell lines. There were more than 1100 upregulated and 2200 downregulated genes (fold change of >1.5 and *p*-adjusted value < 0.05) in Sup-PR cells. When examining only the transcripts with increased levels, gene ontology (GO) analysis demonstrated that there was a strong enrichment of genes involved in metabolic processes (Figure 1A and Appendix A). Of the downregulated genes, the strongest enrichment was found for cell signalling, but phosphorous metabolism (which involves for example nucleotide metabolism) was also highlighted by the analysis (Appendix A). Further gene set enrichment analysis (GSEA) provided more details into which cellular processes may be altered in the prednisolone-resistant line. We found that oxidative phosphorylation and, interestingly, mTORC1 and MYC signalling are likely to be enhanced in Sup-PR cells (Figure 1B and Appendix A).

At this point, we could already see that changes in cell metabolism may be one of the key features of the acquired resistance to prednisolone in our model system. Therefore, we next examined the metabolomes of the two cell lines through polar metabolite profiling by liquid chromatography with tandem mass spectrometry (LC-MS/MS). We first compared the baseline alterations between Sup-B15 and Sup-PR cells. There were a total of 68 metabolites (51 upregulated and 17 downregulated) that were altered by at least 1.2-fold in the GC-resistant line (Figure 1C). Of the 51 compounds detected at higher levels in Sup-PR cells, there were several key players in glycolysis, such as glyceraldehyde-3-phosphate, fructose-6-phosphate, glucose-6-phospate, and α-ketoglutarate, that were at the top of the list (Figure 1C). Metabolite set enrichment analysis (MSEA) demonstrated that glycolysis, gluconeogenesis, galactose and lactose metabolism were, indeed, statistically significantly upregulated (Appendix A). Interestingly, the top hits from the MSEA were related to amino acid regulation—urea cycle, aspartate, glutamate, arginine, proline, phenylalanine and proline metabolism (Appendix A). Importantly, a complementary analysis of the impact that these metabolites have on their relevant pathways (pathway analysis) suggested that the effect on glycolysis of the different sugar-containing compounds that were changed in Sup-PR cells may be less prominent. In contrast, the impact that aberrant amino acid levels have on their respective biosynthesis pathways, such as for aspartate and glutamate metabolism, appeared to be much higher (Figure 1D).

Interestingly, of the 17 metabolites which were downregulated in Sup-PR cells compared to their PR-sensitive parental line, the strongest presence was of nucleotide metabolites (AMP, ADP, UMP and IMP, as well as uridine and cytidine). MSEA and pathway analyses suggested a strong overrepresentation, but a relatively modest impact on purine, pyrimidine and glycerophospholipid synthesis (Appendix A), which was in support of the findings from RNA-seq GO analysis (Appendix A). 

### 2.2. Integrated Transcriptomic and Metabolomic Analysis Reveals the Importance of Glutamine Metabolism in Prednisolone-Resistant Cells

The next question we asked was how the observed transcriptional and metabolomic changes may be related to each other and how they may underpin the acquired resistance in Sup-PR cells. Therefore, we carried out an integrated analysis of the two types of omics data by two independent tools (MetaboAnalyst and MetScape), which investigate the functional connections between genes and compounds. Notably, we found that 22 pathways were significantly enriched. Narrowing down the potential underlying reasons for prednisolone resistance in our in vitro model system to the top 10 pathways, this analysis managed to confirm our previous observations and pinpoint the importance of arginine, alanine, aspartate and glutamate metabolism, as well as glycolysis and pyruvate metabolism (Figure 2A). 

Upon further and more detailed analysis of the exact players involved in these altered pathways, we noticed that there were 15 genes and compounds involved in alanine, aspartate and glutamate metabolism. Interestingly, we found a direct link between upregulated genes and compounds in the glutamine-glutamate-2-oxoglutarate axis—l-glutamine, glutamate dehydrogenase (GLUD1) and oxoglutaric acid (α-ketoglutarate)— which was confirmed by another type of integrated network analysis (Figure 2B and Appendix A). This, together with the recurrent enrichment for glutamine (Gln) metabolism and the high impact of glutamate on fuelling the TCA, both through oxoglutaric acid and pyruvate (the metabolism of which also appeared upregulated), strongly suggested the central role of Gln metabolism in the resistance of Sup-PR cells. Therefore, we hypothesised that targeting the Gln metabolism axis may have therapeutic potential, at least in vitro. 

### 2.3. Targeting Glutamine Metabolism in cALL Cells 

To address the hypothesised significance of Gln metabolism in cALL, we attempted to inhibit this pathway using three different strategies—by Gln starvation (medium without Gln), by competitive inhibition of the Gln transporter ASCT2 with V-9302 [13], and by blockage of glutamate dehydrogenase 1 (GDH or GLUD1) with epigallocatechin gallate (EGCG), which is a natural compound found in green tea. We found that Gln starvation had the strongest impact on apoptosis, leaving only about 20% surviving cells after 3 days of growth in this medium (Figure 3A). Treatment with V-9302 or EGCG also induced apoptosis in both Sup-B15 and Sup-PR cells, but these two drugs showed a more moderate effect on cell viability, even though this may be expected at their IC_50_ concentrations (Figure 3B,C). We did not see a statistically significant difference in the observed apoptosis in PR-sensitive or -resistant cells.

We next examined the potential mode of action of these three therapeutic strategies. Since the Gln metabolism axis can feed into the TCA (through 2-oxoglutarate) and from there into OxPhos (through NADH and FADH), we asked whether there may be changes in mitochondrial respiration and function following incubation for 72 h in these three conditions. Basal respiration, maximal respiration and ATP production were significantly downregulated in all cases to a similar degree in both Sup-B15 and Sup-PR cells, suggesting a similar mode of action between the three treatment types (Figure 4A). 

The mode of action of Gln-free medium can be anticipated, and that of V-9302 has been described previously [13,14,15]. While both of these treatments are very selective against their targets, the effects of EGCG on the metabolome are more unclear and have not been investigated previously. Therefore, we undertook the aforementioned approach with metabolomics analysis to elucidate the way in which EGCG may elicit its anti-leukaemic effect. Upon treatment with this inhibitor, there were 36 detected metabolites that were up- or downregulated in Sup-B15 and PR cells by at least 1.2-fold. Of these, only 4 showed higher abundance in treated cells, whereas the remaining 32 were depleted compared to untreated cells (Figure 4B). These included nucleotides (IMP, UMP and AMP being the most downregulated ones), but also glutamate and 2-oxoglutaric acid (α-ketoglutarate). Pathway analysis and MSEA revealed that EGCG could indeed downregulate Gln and glutamate metabolism (with the highest pathway impact score; Figure 4B and Appendix A). Moreover, we observed that this natural compound could also deplete metabolites involved in purine and pyrimidine metabolism (top hits mentioned above), as well as in arginine (and urea cycle), proline, alanine, aspartate metabolism and others (Figure 4B and Appendix A). The four detected upregulated metabolites—cytidine, creatine, acetylcarnitine and 4-guanidinobutyric acid—did not show significant involvement in any pathway (neither in terms of statistical significance of FDR, nor regarding pathway impact scores—data not shown). 

Lastly, we attempted to validate our results using the only other known pair of GC-sensitive and resistant paediatric ALL cell lines—the MLL-rearranged SEM and SEM-K2 cells [16,17]. While the effect on apoptosis following treatment with the three experimental conditions was weaker than that in Sup-B15 and Sup-PR cells (data not shown), all three treatment strategies managed to elicit a similar mitochondrial function profile, as observed before with reduction in basal and maximal respiration and ATP production (Appendix A). Notably, although Sup-PR and SEM-K2 cells appeared to have different mitochondrial functionality profiles, as measured by mito stress tests (namely lower OCR in Sup-PR cells, as described previously [12]), Gln deprivation and treatment with V-9302 and EGCG showed very consistent results with regard to mitochondrial respiration and ATP production. We also attempted to investigate the molecular changes following Gln metabolism inhibition by EGCG. Even though the two pairs of cell lines had different degrees of prednisolone response (Appendix A—IC_50_) and baseline metabolism (Appendix A), the effect of this drug which we demonstrated in Sup-B15 and Sup-PR cells could also be observed to an extent in SEM and SEM-K2 cells as well. MSEA suggested that, similarly to what we noted in Sup-B15 and PR cells, the urea cycle, alanine and aspartate metabolism were targeted by this drug (Appendix A). Glutamate and arginine metabolism were also in the list of hits, but at a much lower significance score. Therefore, at least some of the findings could be successfully validated in SEM and SEM-K2 cells. 

Importantly, in the context of clinical data, our findings fit well with the only available transcriptomic meta-analysis of matched cALL samples at diagnosis and relapse [10]. By examining all upregulated genes in these datasets, we noticed that certain metabolic processes are enriched in patient samples following treatment failure, including amino acid metabolism (Figure 5A). Interestingly, we also found that there are 142 upregulated genes in Sup-PR cells that are elevated at relapse. Notably, the involvement of these transcripts appeared to be in nitrogen compound, monosaccharide biosynthesis and IMP metabolism (Figure 5B). Lastly, since there has been no metabolomic analysis of patient samples at diagnosis and relapse, we asked how the metabolites we found to be dysregulated in Sup-PR cells would fit into the transcriptomic network of drug-resistant lymphoblasts. Importantly, as observed in Sup-PR cells, this integrated analysis suggested that upon relapse, there is significant enrichment of genes involved in nucleotide biosynthesis, arginine, glutamate and aspartate metabolism, as well as in glycolysis or gluconeogenesis (Figure 5C). Therefore, highlighting these bioenergetic processes, we suggest that they might serve as potential therapeutic targets in vivo.

## 3. Discussion

A lack of response to glucocorticoids (most frequently to prednisolone) is believed to be the main cause of relapse in cALL. However, studying the particular cellular and molecular mechanisms of GC-resistance in vivo/ex vivo is extremely challenging, since GCs are always used in combination with other chemotherapeutic drugs, which will inevitably interfere with any conclusions on the effects of prednisolone as a single agent. To tackle this issue, in the present study, we took advantage of the only two available in vitro models of matched GC-sensitive (Sup-B15 and SEM) and -resistant (Sup-PR and SEM-K2) cell lines. By carrying out integrated transcriptomic and metabolomic analyses, we aimed to elucidate molecular changes that may underpin the lack of response to prednisolone, but also to find novel potential druggable targets in cALL. We demonstrate that at least in some cases alterations in cell metabolism (namely increase in the levels of certain amino acids and glycolysis products) may be a feature of prednisolone-resistant cells. Driven by the concomitant transcriptional and metabolomic signatures suggesting the importance of the glutamine-glutamate-TCA axis for the acquired lack of response to prednisolone in our experimental system, we tested whether the inhibition of different steps in this metabolic pathway would have an anti-leukaemic effect. We addressed the latter question by implementing three different strategies—Gln starvation; treatment with V-9302, which inhibits the ASCT2 Gln transporter; and suppression of glutamate dehydrogenase by EGCG. Functional tests for apoptosis and mitochondrial activity demonstrated the significance of glutaminolysis in both GC-sensitive and -resistant cell lines. L-asparaginase is a critical component of cALL therapy, and is well-known to exert its effects by also deaminating Gln (even though this is not its main target) [18]. Glutamine addiction is detected as a common phenomenon in many types of cancer [19], including T-cell ALL [20]. This is the first report in which drugs targeting Gln metabolism have been effective in prednisolone-resistant cells.

Notably, V-9302 has been described to block Gln uptake very effectively and has been shown to have anti-tumour effects in more than 30 different cell lines (but not in lymphoblastic leukaemias), as well as in mouse breast cancer cell line xenografts [13,21]. This drug has further been tested in liver cancer cells [15] and in colon, breast and lung in vivo models [22]. Its mechanism of action by inhibition of solely ASCT2 has been debated [23], but its ability to block Gln uptake, likely through other amino acid transporters such as SNAT2 and LAT1, has been established by multiple groups [13,14,15,22]. Thus, V-9302 is considered as an excellent tool for investigating Gln deprivation. 

EGCG, on the other hand, has been shown to have more than one target besides glutamate dehydrogenase (GDH/GLUD1). A limitation of this study is the lack of biochemical assays, which would demonstrate the exact mechanism of action of this drug. Several groups, however, have previously characterised the ability of EGCG to inhibit GLUD1 in a wide range of concentrations—from nanomolar range in cell-free assays to 10–100 µM when using pre-treated cells and measuring GLUD1 activity in the cell lysates [24,25,26]. Notably, the anti-cancer properties of this green tea compound exerted by the inhibition of glycolysis have been shown in a number of solid tumours, as well as in myeloid leukaemia [26,27], but this is the first report examining the effect of EGCG on paediatric B-cell ALL. Although the IC_50_ concentrations of this drug used in our study fall within the previously reported effective doses blocking GLUD1, and our data suggest downregulation of glutamate metabolism following treatment with EGCG for 16 h (Figure 4B—downregulation of 2-oxoglutaric acid and MSEA; Appendix A), there may be other effects of this catechin that may contribute to the phenotypes we observed. 

For instance, glucose-6-phosphate dehydrogenase and isocitrate dehydrogenase have also been proven to be hit by EGCG [25,28], while glutaminolysis could be targeted at another level by blocking the glutamate transport GLT-1/EAAT2 (albeit at higher concentrations) [29]. Therefore, the effect of this drug cannot be attributed to a single target. Indeed, in our enrichment and MSEA analyses, we observed that aside from Gln metabolism, gluconeogenesis is also affected (Figure 4). However, there are 18 ongoing clinical trials with EGCG and more than 70 completed ones (based on ClinicalTrials.gov (accessed on 13 January 2023)), making it a relevant candidate for potential clinical blockage of glutaminolysis. 

Importantly, glutamine starvation and blocking of Gln import through V-9302, as well as of glutamate dehydrogenase through EGCG, were effective at disrupting mitochondrial function. They all induced apoptosis in both GC-sensitive and -resistant cell lines at similar levels. However, even though there was an apparent upregulation of Gln metabolism in Sup-PR cells, this cell line was not more susceptible to these three therapeutic approaches compared to Sup-B15 cells (besides a slightly lower IC_50_ value for EGCG). There could be two independent phenomena in prednisolone-sensitive and -resistant cells that may explain the observed results. 

First, regarding the good response to the inhibition of glutaminolysis in Sup-B15 and SEM cells, the central role of Gln in nucleotide, lipid, non-essential amino acid biosynthesis and others [30] has long been highlighted as a therapeutic target in cancer in general [19,30], including haematological malignancies (AML) [31,32], but not cALL in particular. The only relevant studies in this paediatric haemato-oncologic malignancy are in the context of L-asparaginase therapy, which is well-known to also deaminate Gln to glutamate (i.e., having glutaminase activity). In this context, however, the extra effect of L-asparaginase is believed to be beneficial not only because it blocks the anaplerotic role of Gln in nucleotide biosynthesis [33], but mostly because it prevents this amino acid from becoming a nitrogen donor for asparagine production in the cell [34], thereby depleting both external and internal pools of asparagine. Therefore, the response of prednisolone-sensitive cells to all three experimental conditions in this study may be demonstrating an unsurprising Gln addiction in cALL, which has not been described nor well-characterised to this point. 

Second, with regards to prednisolone-resistant cells, besides Gln metabolism, there were several other transcriptional and biosynthesis pathways that were altered in Sup-PR cells. We observed enrichment of aspartate, arginine, alanine, phenylalanine and other amino acid metabolisms, which can likewise fuel nucleotide biosynthesis and the TCA (e.g., asparagine and alanine can be converted to oxaloacetate and pyruvate, respectively). Thus, it is possible that our GC-resistant model cells may be rewired for greater metabolic plasticity, which may allow them to compensate for the block of Gln uptake or production of 2-oxoglutarate. Therefore, targeting Gln metabolism alone may not be sufficient to induce apoptosis in all cells, and a combination with another inhibitor may be more beneficial. 

Importantly, another potential rescue mechanism in Sup-PR cells, aside from the proposed metabolic plasticity, which holds a considerable therapeutic potential is the activation of two signalling pathways—mTORC1 and MYC—as is apparent from the GSEA (Appendix A). The mammalian target of rapamycin (mTOR) signalling is deregulated in multiple malignancies, and there is a constant feedback loop between amino acid levels and the activation of this pathway [35,36]. Several amino acids, including Gln, can serve as signalling molecules to activate mTORC1 [37], which, in turn, can control mitochondrial function (the TCA), nutrient uptake and glutaminolysis [38] (by upregulation of the glutamate transporter EAAT1 in T-ALL [39]), while also ultimately enhancing cell growth and survival [40]. Notably, the involvement of the mTORC1 pathways in lymphoblastic leukaemias and chemoresistance has been suggested previously [41]. Targeting this complex has already shown potential in a high risk B-ALL xenograft model [42] and in combination with Gln depletion in Notch1-driven T-ALL [20]. The role of MYC in controlling leukaemic cell metabolism [43] and Gln catabolism has been well-established as well [44]. In MLL-rearranged cALL xenografts, this oncoprotein has been also shown to be able to regulate both OxPhos and glycolysis, rendering lymphoblasts a context-specific metabolic plasticity related to the leukaemia-initiating cell state [45]. Thereby, the activation of MYC signalling in Sup-PR cells may provide another metabolic fitness route that could explain why a sub-population of these GC-resistant cells is not susceptible to V-9302, EGCG or glutamine deprivation. Therefore, while targeting glutamine metabolism in prednisolone-resistant cells does show efficacy, the anti-leukaemic potential of this approach will perhaps be enhanced if combined with inhibition of other amino acid biosynthesis pathways (such as asparaginase) or of signalling axes such as the ones controlled by mTORC1 or c-MYC. A good example of the promising potential of combinatorial treatments is provided by the co-targeting of glycolysis with 2-deoxyglucose, and of glutaminolysis with V-9302. This approach showed greater efficacy in breast, colon and lung cancer cell lines [22]. Simultaneous blocking of glutaminase by CB-839, and of Gln import with V-9302, has shown synergistic effects in hepatocellular carcinomas [15]. The same ASCT2 transporter inhibitor has also been successfully used to resensitise Paclitaxel-resistant ovarian cancer cells [46]. Targeting glutaminase with CB-839 has also shown synergy with Bruton tyrosine kinase inhibition in ibrutinib-resistant mantle B-cell lymphoma cells [47]. With regard to the other drug used in our study, EGCG, with its multiple potential molecular targets, has been demonstrated to reverse resistance to cisplatin [48] and to the tyrosine kinase inhibitor gefitinib [49] in lung cancer cells, as well as to 5-fluor uracil in colorectal cancer cells [50]. In that context, our study could provide strong rationale for testing a number of different chemotherapeutic agents, together with inhibitors of glutaminolysis, in both prednisolone-sensitive and -resistant childhood B-cell ALL cells.

In conclusion, in this study, we investigated the transcriptional and metabolomic profile of prednisolone-resistant cALL cells. The novel data we present herein demonstrate that the acquired lack of sensitivity to prednisolone is accompanied by gross metabolic rewiring towards amino acid biosynthesis and glycolysis. Comparison of our data to a publicly available meta-analysis suggested that similar events may also occur upon relapse. Among other druggable and potentially dysregulated metabolic or signalling pathways, glutaminolysis presents an attractive therapeutic target. We demonstrate that blocking Gln availability or its conversion to 2-oxoglutarate impairs mitochondrial function and induces apoptosis in both GC-sensitive and -resistant cells. Therefore, this study provides not only novel insights into the mechanisms of resistance to prednisolone, but also a basis for further investigations, including potential combinatorial treatments aiming to tackle resistance to prednisolone, asparaginase and other chemotherapeutic agents. 

## 4. Materials and Methods

### 4.1. Cell Culture, Drug Titrations, Cell Proliferation and Apoptosis Assays

Sup-B15 (DSMZ, Braunschweig, Germany) and Sup-PR (developed in our laboratory as described previously) [12] cell lines were cultured in IMDM with 20% FBS and 0.5% Pen/Strep (PAN-Biotech, Aidenbach, Germany). SEM (DSMZ, Braunschweig, Germany) and SEM-K2 (kindly provided by Dr. Tino Schenk from the University Hospital of Jena (Germany)) cell lines were grown in IMDM with 10% FBS and 0.5% Pen/Strep (PAN-Biotech, Aidenbach, Germany). For Gln starvation experiments, IMDM without Gln (PAN-Biotech, Aidenbach, Germany) was used. EGCG and V-9302 were purchased from Cayman Chemical, Ann Arbor, MI, USA (European Division in Estonia) and dissolved in DMSO (PAN-Biotech, Germany). Drug titrations were carried out for 72 h, with at least 4 different concentrations of each drug in biological duplicates and in technical duplicates or triplicates. Cell viability was measured using a standard MTT protocol. IC_50_ values were determined by log-transforming the drug concentrations, normalising the data to untreated control values and applying a nonlinear regression algorithm for curve fitting using GraphPad Prism 9.4.0. For assessment of apoptosis following Gln starvation or drug treatments for 72 h, an Annexin V/PI kit (Luminex, Austin, TX, USA) was used on a Guava^®^ Muse^®^ Cell Analyser (Luminex, Austin, TX, USA) instrument.

### 4.2. RNA Sequencing and Data Analysis

Sup-B15 and Sup-PR cells were grown as described above, then 1 million cells of each line were collected and RNA was extracted using a Qiagen RNeasy kit (Qiagen, Germantown, MD, USA), following the protocol provided by the manufacturer. RNA was quantified with Nanodrop (Thermo, Waltham, MA, USA), and samples were sent to Novogene (UK) for further processing—including quality control, library preparation with NEBNext^®^ UltraTM RNA Library Prep Kit for Illumina^®^ (NEB, Ipswich, MA, USA) and pair-end sequencing at >20 M reads/sample on an Illumina sequencer. Data processing was also carried out by Novogene using HISAT2 for mapping and DESeq2 and EdgeR for generating a list with differentially expressed genes (DEGs). DEGs in Sup-PR compared to Sup-B15 cells were filtered for fold change (FC > 1.5) and statistical significance (*p*-value < 0.05). The resulting sub-list was further processed by the BiNGO [51] plugin for Cytoscape [52] for gene ontology (GO) analysis, and by the Broad Institute’s GSEA software [53] for gene set enrichment analysis (GSEA) as previously described [12]. 

### 4.3. Metabolomic Analysis

Cells were seeded at 8 × 10^5^–1 × 10^6^/mL in 12-well plates in biological duplicates, with or without IC_50_ concentrations of EGCG, and incubated for 16 h. Metabolites were extracted by adding 1 mL of ice-cold 80% MS-grade methanol and 3 cycles of snap freezing on dry ice, thawing on ice and vortexing for 40 s. Cell debris were then pelleted (15,000× *g* for 10 min at 4 °C), and the supernatant containing extracted metabolites was transferred to new 1.5 mL Eppendorf tubes and shipped on dry ice for analysis. Liquid chromatography with tandem mass spectrometry (LC-MS/MS) of Sup-B15, Sup-PR, SEM and SEM-K2 cells was carried out by MsOmics (Denmark) as described previously [54]. A Vanquish LC coupled to Thermo Q Exactive HF (Thermo Scientific) with an electrospray ionization interface as the ionization source were used. Compound Discoverer 3.3 (Thermo Scientific) was employed to extract peak areas. Four levels of identification of compounds were applied based on 3 criteria—retention times (compared against in-house authentic standards), accurate mass (with an accepted deviation of 3 ppm) and MS/MS spectra. Compounds identified at level 1 (identified based on all 3 criteria), 2a and 2b (based on 2 of the 3 criteria) were used for further analysis, while compounds at Level 3, based on accurate mass, isotope pattern and reference count, (231 compounds in total) were excluded from the analysis, as this was the annotation method carrying the lowest confidence.

A total of 111 metabolites, detected at Level 1, 2a and 2b, were then filtered to remove likely contaminants such as HEPES, mercaptoethanol, benzylformamide and others. We used a less stringent approach with a log_2_ fold change of peak area values normalised to GC-sensitive or untreated control cells of >1.2 and <0.8 (as applied previously by Sun et al. [55] and by Feng et al. [56]). Thus, we obtained 68 metabolites when comparing Sup-PR to Sup-B15 cells, 42 metabolites when comparing SEM-K2 to SEM cells, 36 metabolites for Sup-B15 and Sup-PR cells treated with EGCG compared to control Sup-B15 and Sup-PR cells and 51 metabolites for the latter comparison in SEM and SEM-K2 cells. These datasets were then applied for further analysis by MetaboAnalyst 5.0 [57]. 

For metabolite set enrichment and for pathway analysis in MetaboAnalyst 5.0, we used default settings and, respectively, the SMPDB database [58] and hypergeometric test for enrichment and for the KEGG pathway library. This was conducted for both pairs of sensitive and resistant cells—Sup-B15 and Sup-PR and SEM and SEM-K2. For network analysis integrating genes and metabolites, we applied fold change values for both types of data (1.5 FC for gene expression and 1.2 fold change for metabolites) as well, and used the full lists of up- and downregulated genes and metabolites. We then selected gene–metabolite interaction network for the network analysis option (again, with default settings for degree and betweenness filters). The statistical analysis was based on KEGG pathways, but the gene–compound network was also run against the Reactome and biological processes databases. This type of integrated gene expression and metabolomics analysis was carried out with MetScape in CytoScape [59] as well (again, using default settings). The same workflow was applied for the analysis of the effect of EGCG. Only enrichment results with statistical significance with *p*-value < 0.05 (or FDR < 0.05 when calculated) were considered for discussion and further assessment.

### 4.4. Seahorse Mitochondrial Function Analysis

Seahorse Mito Stress tests were carried out on a Seahorse XFp Analyzer (Agilent, Santa Clara, CA, USA) following the recommended protocols by the manufacturer and as described previously for Sup-B15 and Sup-PR cells (normalising the readings to cell numbers per well) [12]. In brief, the oxygen consumption rate (OCR) of EGCG-treated or control untreated cells was measured, and then the data were analysed by Wave Software version 2.6.3.5 (Agilent, Santa Clara, CA, USA to generate reports for basal and maximal respiration, as well as for ATP production. The experiments were carried out at least in biological and technical duplicates. For data presentation and statistical analysis (Student’s *t*-test), GraphPad Prism 9 (GraphPad Software, San Diego, CA, USA) was used.

## Figures and Tables

**Figure 1 ijms-24-03378-f001:**
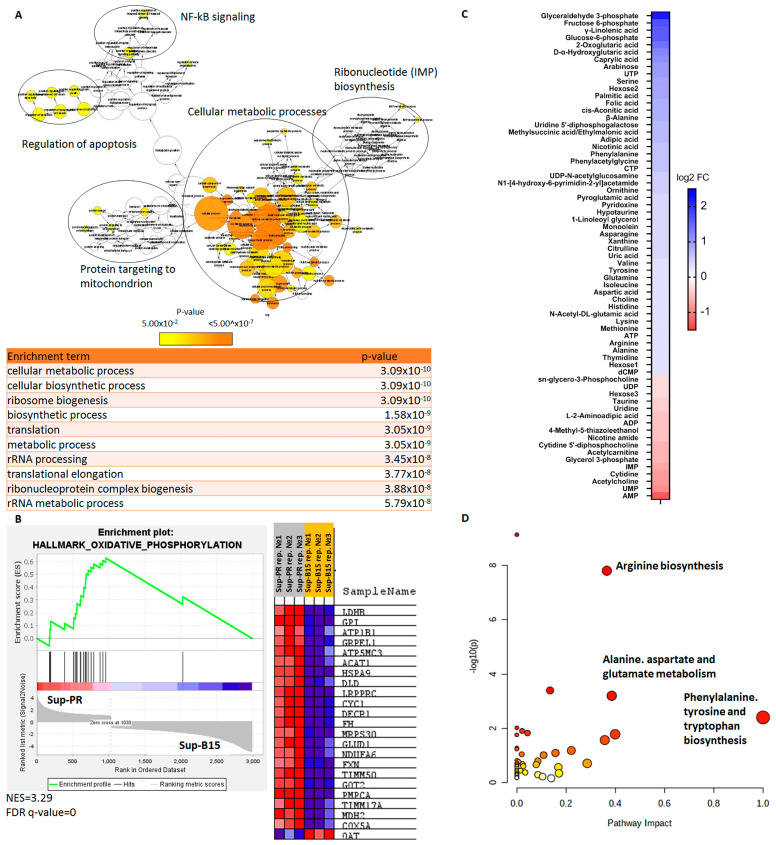
Gene expression and metabolite analyses of prednisolone-resistant cells suggest dysregulated metabolism. (**A**) Gene ontology analysis of upregulated genes in Sup-PR showing overrepresentation of transcripts in several clusters, as annotated, and the statistical significance of the enrichment terms. (**B**) GSEA of differentially expressed genes in Sup-PR cells demonstrating enhanced oxidative phosphorylation and the list of up- (in red) and downregulated (in blue) genes involved in this process (NES—normalised enrichment score, FDR—false discovery rate). (**C**) Heatmap based on log2 fold change, as annotated, of metabolites that are up- or downregulated (in blue and red, respectively) in Sup-PR compared to Sup-B15 cells. (**D**) Metabolite pathway analysis presenting the involvement of the metabolites from (**C**) in biosynthetic pathways, as annotated, showing the statistical significance of the enrichment and the impact they have on the relative pathway.

**Figure 2 ijms-24-03378-f002:**
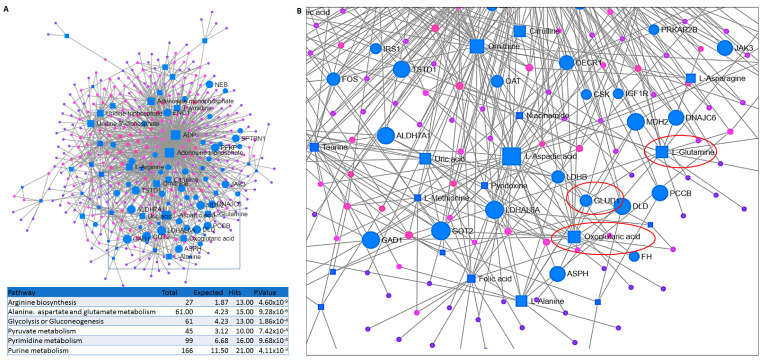
Integrated transcriptomic and metabolomic analysis reveals alterations in the number of biosynthesis pathways. (**A**) Network analysis of differentially expressed genes and metabolites in Sup-PR compared to Sup-B15 cells and statistical data for six of the top ten enriched KEGG pathways. The genes highlighted in blue colour (circles) and metabolites (squares) on the diagram are from the selected pathways. (**B**) A more detailed view of genes and metabolites involved in glutamine metabolism (in red circles)—GLUD1, l-Glutamine and oxoglutaric acid.

**Figure 3 ijms-24-03378-f003:**
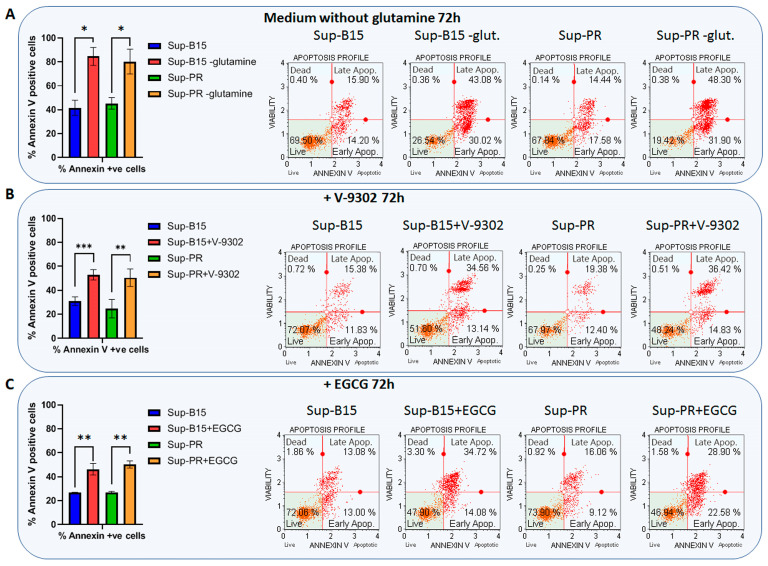
Targeting glutamine metabolism induces apoptosis in cALL cell lines. The percentage of live and annexin-positive cells, and representative flow cytometry plots, of Sup-B15 and Sup-PR cells grown for 72 h in a medium without Gln (top panel), or treated with IC50 concentrations of the Gln transporter inhibitor V-9302 (middle panel) and with the glutamate dehydrogenease inhibitor EGCG (bottom panel). Error bars represent the mean with SD from biological duplicates. *—*p* < 0.05, **—*p* < 0.005, ***—*p* < 0.0005, Student’s *t*-test.

**Figure 4 ijms-24-03378-f004:**
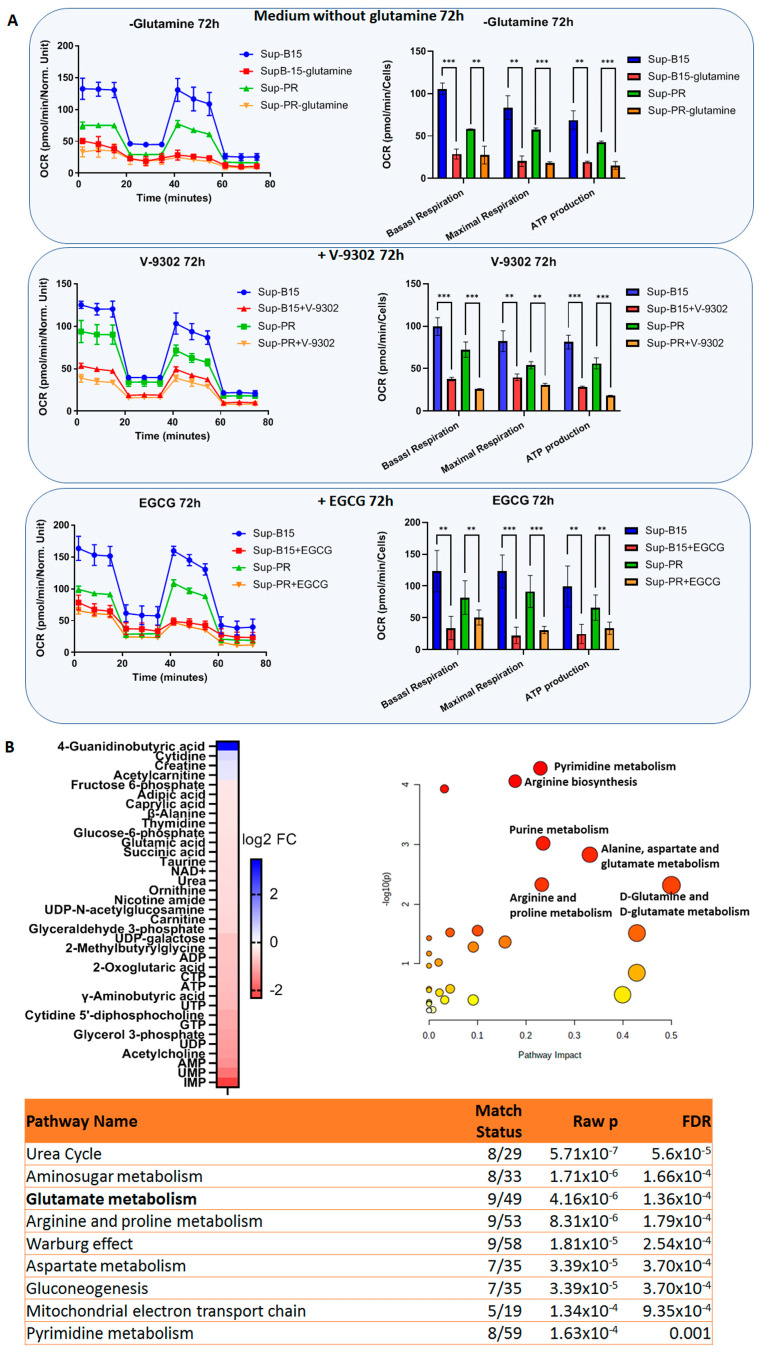
Inhibition of glutaminolysis impairs mitochondrial function. (**A**) Seahorse analysis of mitochondrial function showing oxygen consumption rates (OCR) in a Seahorse mito stress test (left hand side), and the relevant quantification (right hand side) of basal and maximal respiration and ATP production, as annotated in cells grown in a medium without glutamine (top panel), or treated with V-9302 (middle panel) or EGCG (bottom panel) for 72 h. Error bars represent the mean with SD from biological duplicates. **—*p* < 0.005, ***—*p* < 0.005 Student’s *t*-test. (**B**) Metabolite analysis of Sup-PR and Sup-B15 cells treated with EGCG compared to control untreated samples of the two cell lines, showing a heatmap of the up- and downregulated metabolites after treatment for 16 h (left hand side), as well as pathway analysis (right hand side) and a list of the top enrichment terms from a MSEA (bottom).

**Figure 5 ijms-24-03378-f005:**
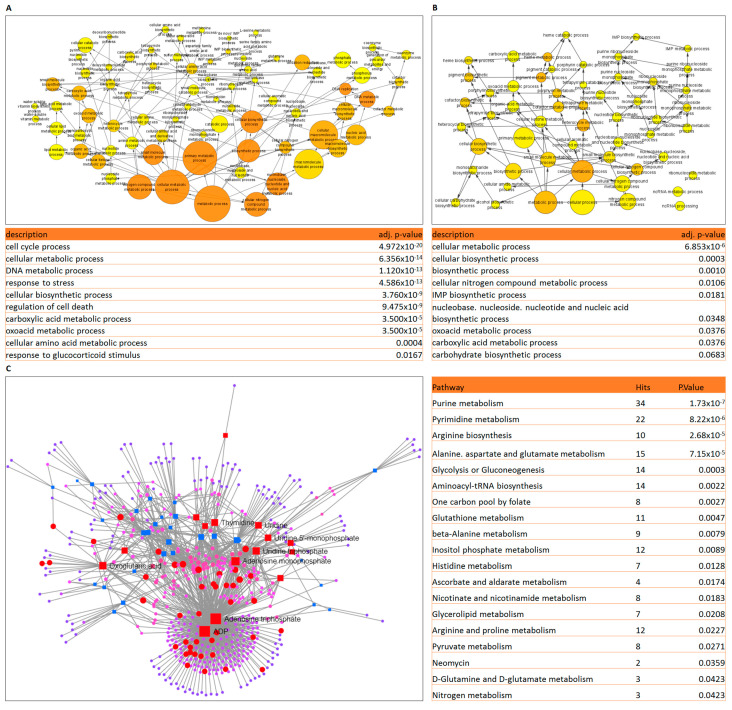
Gene expression analysis of metadata of matched patient samples at diagnosis and relapse. (**A**) GO analysis of upregulated genes in patient samples from meta-analysis [10] showing a metabolism-enriched cluster of nodes from upregulated genes in relapsed cALL patients and selected enrichment terms. (**B**) GO analysis of 142 upregulated genes found in both the meta-analysis and Sup-PR cells showing enrichment for metabolic processes, as annotated. (**C**) Integrated analysis of all up- and downregulated genes in relapse patients [10] and all dysregulated metabolites in Sup-PR cells. The genes (circles) and metabolites (squares) from the top 5 enriched KEGG terms are highlighted in red.

## Data Availability

The RNA-seq raw and processed data can be found in the Gene Expression Omnibus, GEO (https://www.ncbi.nlm.nih.gov/geo/ (accessed on 13 January 2023)) under accession number GSE217428.

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
