# Peer review of "Targeting Glutaminolysis Shows Efficacy in Both Prednisolone-Sensitive and in Metabolically Rewired Prednisolone-Resistant B-Cell Childhood Acute Lymphoblastic Leukaemia Cells"

_ijms, 2023, doi:10.3390/ijms24043378_

Round 1

Reviewer 1 Report

Overall, it seems that this study is new and novel. This paper was well-designed. Anyway, the author can find some comments in the following.

In Line 20, please replaces therapy failure with treatment failure, which is more common.

Please rewrite lines 20 and 21.

In fig 2B, please clarify gray and yellow Sup-PR cells 1, 2, and 3; this figure is unclear; please use the larger size and define all the groups in the caption.

In fig 2C, the annotated proteins are also not visible enough.

Why did they estimate the IC50 just for 72h? Did they have any data for 24 or 48 h?

In fig 2 B, please mention all the metabolites in the circles, such as glutamate dehydrogenase (GLUD1).  

Please discuss if there was any alteration in proteins and metabolites involved in redox homeostasis and oxidative stress in prednisolone-resistant and -sensitive cell lines in the RNAseq and metabolomics analysis.

Author Response

We thank the reviewer for their insightful and constructive comments. Please find below point-by-point responses to the issues raised by the reviewer

  1. In Line 20, please replaces therapy failure with treatment failure, which is more common.

“Therapy failure” has been replaced with “treatment failure” in line 20, 61, 78 and line 271.

  1. Please rewrite lines 20 and 21.

This is now taken care of.

  1. In fig 1B, please clarify gray and yellow Sup-PR cells 1, 2, and 3; this figure is unclear; please use the larger size and define all the groups in the caption.

Figures 1A and B have been made bigger and the grey and yellow labels in 1B have been improved in terms of resolution and clarity.

  1. In fig 1C, the annotated proteins are also not visible enough.

Figure 1C has also been made bigger and the font of the labeled proteins has been increased

  1. Why did they estimate the IC50 just for 72h? Did they have any data for 24 or 48 h?

We have not measured cell proliferation and viability at earlier time points because 1) the cell lines used here have a relatively slow doubling time of ~48-60h and effects on the cell cycle wouldn’t be very visible before the second division (effect on apoptosis, however, would be visible), and 2) the expected mechanism of action of glutamine depletion may not necessarily have an immediate effect on cell proliferation and viability (e.g. there can be different sources of glutamine that may initially compensate for the drug treatment).

  1. In fig 2 B, please mention all the metabolites in the circles, such as glutamate dehydrogenase (GLUD1).  

This is now explained in the figure legend.

  1. Please discuss if there was any alteration in proteins and metabolites involved in redox homeostasis and oxidative stress in prednisolone-resistant and -sensitive cell lines in the RNAseq and metabolomics analysis.

We have specifically searched our RNA-seq analysis for redox homeostasis and oxidative stress related processes and found nothing in the GSEA analysis, and only one hit (“oxidation reduction”) at p-value of 0.0491 from the GO analysis of upregulated genes in Sup-PR cells. With regards to the metabolomics and integrated analyses, the only evidence of potential changes in redox homeostasis come from enrichment for “Glutathione metabolism” at p-value of 0,013 (this is hit number 15 from the table on Figure 2A) and “Glutathione synthesis and recycling”, which, however, has and FDR of 0,328 (Metaboanalyst analysis against REACTOME). Furthermore, in the Mass spec analysis we did not detect glutathione at any level and only had L-g-Glutamyl-L-cysteine at level 3, which is not significant. Even though it is likely that by altering glutaminolysis, the synthesis of glutathione is also going to be changed and redox homeostasis would be affected, we have not looked at ROS levels for examples and feel that given the insufficient amount of data, we cannot include much on this subject in the discussion section of the manuscript.  

Reviewer 2 Report

In this manuscript, Sbirkov et al. present molecular differences (at transcriptomic and metabolomic level) in a prednisolone sensitive cell line (Sup-B15) and their corresponding cell line resistant to prednisolone (Sup-PR), both derived from B-cell acute lymphoblastic leukemia (ALL). Several pathways seems to be altered, however, the authors are committed to modulating the glutamine-glutamate-ketoglutarate axis; finding that inhibition of glutamine metabolism decreases cell viability, oxygen consumption rates and ATP production, and increases apoptosis, both in prednisolone-sensitive cell lines (Sup-B15 and SEM) and prednisolone-resistant cells (Sup-PR and SEM-K2).

Major comments:

Although the central idea is interesting and the title is quite attractive, the experimental strategy is poor, especially to addres the central idea of the title "Targeting glutaminolysis in glucocorticoid-resistant". The authors generalize many of their results, speaking indiscriminately of glucocorticoids and chilhood leukemias when in fact they only present data with prednisolone and leukemia B models, considering the biological differences between pediatric T- and B-cell ALL (doi: 10.1016/S1470-2045(19)30031-2).

Methods.

The used GCG concentration was chosen based on a dose-response curve of only 3 points (3 concentrations; Supplementary figure 4, bottom panel). In addition, the effect evaluated was cell viability (MTT assay) rather than determining the concentration that effectively inhibits GDH; thus, it cannot be ensured that the effect observed on viability, death and OCR ir a consequence of the inhibition of the enzyme. In this sense, reference 24 mentions what up to 20 uM would be sufficient for the inhibition.

In several experiments, only biological duplicates are made, is this enough to consider a statistically significant result?

Results.

The result presented in the Figures 3, 4 (up panel) and supplementary figure 5, seem to indicate that the strategy against glutamine metabolism would not have a different effect on cell lines sensitive and resistant to prednisolone, which is mentioned in line 192 and the discussion (lines 325; 341-344), but it is still contradictory to what has been stated in the title and general intent of the study. 

Figure 3: how do you explain the decresed viability in the controls (blue and green bars)? wher even in the presence of glutamine in the medium and without the addition of V-9302 and EGCG there are only 60-70% of viable cells. Is this normal for these cell lines?

Figure 4A: Is there any explanation for the difference in basal OCR levels between Sup-B15 and Sup-PR? which are not so pronounced in SEM/SEM-K2

Evaluating the effect of EGCG and prednisolone together could yield interesting data, especially if it allows to overcome resistance in Sup-PR and SEM-K2 cell lines. Otherwise, this work could be limited only to the transcriptomic and metabolomic study of prednisolone-sensitive and resistant cell lines, with an adequate discussion that opens the panorama to new studies. In this sense, the discussion should be expanded an improved.

Minor coments:

The changes at the level of the metabolites found in the cell lines Sup-B15 and Sup-PR are interesting; however, some metabolites are both upregulated and downregulated (related to lactose synthesis, alanine and pyrimidine metabolism). What is this due to?

Line 222-223: "action of glutamine-freemedium can be anticipated and the one of V-9302 has been described before". Where? Should be mentioned

Line 19: acute lympoblastic leukaemia should be corrected "acute lymphoblastic leukemia"

Figures

Figure 3: homogenize figure legends. In top panel, green bars is are labeled as Sup-PR while in middle and bottom panel only "PR"; same for orange bars. In dot plots the text of the gates is above, it is no clear if it is "dead late", "apop/dead" or "late apop/dead". Titles of dot plots are the same in the three panels (Sup-B15-glut or Sup-PR-glut). In footnote is neccesary to mention the value that is considered for ***, since that level of significance is found in the graphs

Figure 4A, righ graph: homogenize figure legends in bottom panel (PR instead Sup-PR); middel panel: Sub-B15+EGCG should be Sub-B15+v-9302, same for Sup-PR+EGCG

Supplemenraty figure 4, top panel: How are negative levels of viability explained? it is correct inhibitor concentration on the X axis?

Author Response

We are very grateful to Reviewer 2 for their detailed work and justified and constructive comments.

Major comments:

  1. Although the central idea is interesting and the title is quite attractive, the experimental strategy is poor, especially to address the central idea of the title "Targeting glutaminolysis in glucocorticoid-resistant". The authors generalize many of their results, speaking indiscriminately of glucocorticoids and chilhood leukemias when in fact they only present data with prednisolone and leukemia B models, considering the biological differences between pediatric T- and B-cell ALL (doi: 1016/S1470-2045(19)30031-2).

In relation to another 2 points raised below, the title has been modified in attempt to present more accurately the findings in the paper. - Targeting glutaminolysis shows efficacy in both prednisolone-sensitive and in metabolically rewired prednisolone-resistant B-cell childhood acute lymphoblastic leukaemia cells

Regarding glucocorticoids, we agree that dexamethasone and prednisone (and its active form prednisolone) indeed have differences and that we cannot generalize our experimental system and results to dexamethasone since we have not tested it on our cell lines. Therefore,  we have attempted to clarify and discriminate between GCs by explicitly saying that we refer to prednisolone in lines 293, 306, 316, 357, 369, 403, and 423.

A clear discrimination has been made now throughout the text between T- and B-cell ALL (e.g. line 56 and 59), and most importantly in the title, which places the whole manuscript in the right context - Targeting glutaminolysis shows efficacy in both prednisolone-sensitive and in metabolically rewired prednisolone-resistant B-cell childhood acute lymphoblastic leukaemia cells

Methods.

  1. The used GCG concentration was chosen based on a dose-response curve of only 3 points (3 concentrations; Supplementary figure 4, bottom panel). In addition, the effect evaluated was cell viability (MTT assay) rather than determining the concentration that effectively inhibits GDH; thus, it cannot be ensured that the effect observed on viability, death and OCR ir a consequence of the inhibition of the enzyme. In this sense, reference 24 mentions what up to 20 uM would be sufficient for the inhibition.

The inhibition of GDH by EGCG (and the potential block of other enzymes and processes) and the limitation of our work in that respect are now discussed further in a new paragraph - line 320-341.

We agree that the manuscript would have been stronger if we had examined the inhibition of GDH in our in vitro model too, however, this type of assays were beyond the scope of this work. Nevertheless, indirect evidence of the inhibition of GDH (how much and is it sufficient is difficult to determine) comes from the metabolomic data where a-ketoglutarate is downregulated 16 hours after treatment with EGCG. Also, the concentrations of EGCG we used fall within the previously published concentrations (10, 20, 30 and 100uM) that would inhibit GDH in several different model systems. Therefore, we are most likely capable of inhibiting GDH too, but it is also possible that EGCG inhibits other enzymes and processes at these concentrations, which is also discussed in the text.

  1. In several experiments, only biological duplicates are made, is this enough to consider a statistically significant result?

Biological duplicates in technical duplicates or triplicates presenting 4-6 data points per group are sufficient for statistical analysis and determination of the significance of the result (i.e. measurement of the variance). Importantly, most of the experiments are reproduced in SEM and SEM-K2 cells, which is in support of the biological significance of the results too.

Results.

  1. The result presented in the Figures 3, 4 (up panel) and supplementary figure 5, seem to indicate that the strategy against glutamine metabolism would not have a different effect on cell lines sensitive and resistant to prednisolone, which is mentioned in line 192 and the discussion (lines 325; 341-344), but it is still contradictory to what has been stated in the title and general intent of the study. 

The initial hypothesis was that prednisolone-resistant cells would be more sensitive to targeting glutaminolysis than prednisolone-responsive cells. A large part of the discussion addresses the potential reasons why this strategy may be effective in both settings – line 358-402. We have now changed the title to represent more accurately the final results and conclusions of the study. - Targeting glutaminolysis shows efficacy in both prednisolone-sensitive and in metabolically rewired prednisolone-resistant B-cell childhood acute lymphoblastic leukaemia cells. Ultimately, even if not more sensitive to this strategy, we believe that it is still of interest that prednisolone-resistant cells are sensitive to it. 

  1. Figure 3: how do you explain the decreased viability in the controls (blue and green bars)? wher even in the presence of glutamine in the medium and without the addition of V-9302 and EGCG there are only 60-70% of viable cells. Is this normal for these cell lines?

This decrease in cell viability is an interesting observation that we have actually seen before as well when setting up experiments in 12 or 24-well plates. Sup-B15 and Sup-PR cells in general are not easy to grow well, which is also suggested by the requirement of 20% FBS to stimulate them,and there would normally be around 15% dead cells. It is possible that this lower than usual viability (with further ~15%) in these rather sensitive cells is related to moving the cells from a flask to 24-well plates for the experiments i.e. placing the cells into very different conditions - different surface to volume ratios in the plates (e.g. >1cm of medium on top of them), different aeration in the plate etc. Nevertheless, the induction of apoptosis following treatment with EGCG and V-9302 (and in medium without glutamine) is substantial and significant and we have decided to focus on this by changing the graphs – now the % annexin V positive cells are normalized to the untreated controls to show the enhanced apoptosis in the three experimental conditions.  

  1. Figure 4A: Is there any explanation for the difference in basal OCR levels between Sup-B15 and Sup-PR? which are not so pronounced in SEM/SEM-K2

Yes, we have previously described this difference in OCR between Sup-B15 and Sup-PR cells ( doi: 10.3389/fonc.2021.632181 ). However, we have not focused on SEM and SEM-K2 cells before. As shown in supplementary figure 6, Sup-PR and SEM-K2 appear to have distinct metabolism so OxPhos is likely to be different too in the two sets of cell lines. This is now mentioned in the relevant results section – line 254-258.

  1. Evaluating the effect of EGCG and prednisolone together could yield interesting data, especially if it allows to overcome resistance in Sup-PR and SEM-K2 cell lines. Otherwise, this work could be limited only to the transcriptomic and metabolomic study of prednisolone-sensitive and resistant cell lines, with an adequate discussion that opens the panorama to new studies. In this sense, the discussion should be expanded an improved.

We agree that combinatorial treatments have much greater potential than single agent therapy and that re-sensitisation to prednisolone is a very attractive goal. We have already touched on that subject in line 379-380 and 403-404, but this is now expanded as recommended – lines 406-420 and 431.

Minor coments:

  1. The changes at the level of the metabolites found in the cell lines Sup-B15 and Sup-PR are interesting; however, some metabolites are both upregulated and downregulated (related to lactose synthesis, alanine and pyrimidine metabolism). What is this due to?

This is another interesting observation made by Reviewer 2 –e.g. UTP is upregulated and UMP is downregulated. There is data in literature regarding the ratios of tri- to di-phosphate pyrimidines in relation to cancer and mitochondrial function. However, given the multiple possible pathways of dysregulating certain metabolites and the complex interconnections between different metabolic pathways, we feel that it would be too speculative to discuss these further in the manuscript.

  1. Line 222-223: "action of glutamine-freemedium can be anticipated and the one of V-9302 has been described before". Where? Should be mentioned

The relevant references are now added.

  1. Line 19: acute lympoblastic leukaemia should be corrected "acute lymphoblastic leukemia"

This is now corrected.

Figures

  1. Figure 3: homogenize figure legends. In top panel, green bars is are labeled as Sup-PR while in middle and bottom panel only "PR"; same for orange bars. In dot plots the text of the gates is above, it is no clear if it is "dead late", "apop/dead" or "late apop/dead". Titles of dot plots are the same in the three panels (Sup-B15-glut or Sup-PR-glut). In footnote is neccesary to mention the value that is considered for ***, since that level of significance is found in the graphs

This is taken care of now.

  1. Figure 4A, righ graph: homogenize figure legends in bottom panel (PR instead Sup-PR); middel panel: Sub-B15+EGCG should be Sub-B15+v-9302, same for Sup-PR+EGCG

This is taken care of now.

  1. Supplemenraty figure 4, top panel: How are negative levels of viability explained? it is correct inhibitor concentration on the X axis?

Negative values resulted from subtracting the background absorption (wells without cells) from the reading of the treated wells with cells.The lowest negative value is -0.041 while the highest positive values is ~0.370. Therefore, we do not see anything disturbing in going below the X axis with about 1% of the absorption maximum.  

The titles of X axes are now changed to µM prednisolone/EGCG/V-9302 concentration (log).

Round 2

Reviewer 2 Report

I am grateful to the authors for their attention to comments made; the manuscript has now substantially improved and there ir congruence between the main idea and the results achieved, despite the limitations of the methods.